ecology, cellular biology

coral reefs, mitochondria, unfolded protein response, climate change, stress response

**Author for correspondence:**
Mark W. Pellegrino
e-mail: mark.pellegrino@uta.edu

# Uncovering a mitochondrial unfolded protein response in corals and its role in adapting to a changing world

Bradford A. Dimos, Siraje A. Mahmud, Lauren E. Fuess, Laura D. Mydlarz and Mark W. Pellegrino

Department of Biology, University of Texas at Arlington, Arlington, TX 76019, USA

BAD, 0000-0003-0899-6378

The Anthropocene will be characterized by increased environmental disturbances, leading to the survival of stress-tolerant organisms, particularly in the oceans, where novel marine diseases and elevated temperatures are re-shaping ecosystems. These environmental changes underscore the importance of identifying mechanisms which promote stress tolerance in ecologically important non-model species such as reef-building corals. Mitochondria are central regulators of cellular stress and have dedicated recovery pathways including the mitochondrial unfolded protein response, which increases the transcription of protective genes promoting protein homeostasis, free radical detoxification and innate immunity. In this investigation, we identify a mitochondrial unfolded protein response in the endangered Caribbean coral *Orbicella faveolata*, by performing *in vivo* functional replacement using a transcription factor (Of-ATF5) originating from a coral in the model organism *Caenorhabditis elegans*. In addition, we use RNA-seq network analysis and transcription factor-binding predictions to identify a transcriptional network of genes likely to be regulated by Of-ATF5 which is induced during the immune challenge and temperature stress. Overall, our findings uncover a conserved cellular pathway which may promote the ability of reef-building corals to survive increasing levels of environmental stress.

## 1. Introduction

Coral reefs have recently experienced massive declines [1,2], primarily driven by marine diseases [3,4] and thermally induced mass coral bleaching [5,6]. As a result, many studies have investigated the mechanisms with which corals respond to disease [7–9] as well as factors associated with bleaching [10–12]. Interestingly, these pathways show considerable overlap [13], and involve both antioxidants [14,15] and molecular chaperones, which have been suggested to mediate a protective response [16]; however, additional cellular pathways promoting these protective responses are likely to exist.

Both bacterial toxins and thermal stress lead to mitochondrial dysfunction [17,18], suggesting a possible common mechanism worth further investigation. Cells use a variety of means to mitigate dysfunction to mitochondria, including the mitochondrial unfolded protein response (UPR^mt), a retrograde pathway which functions to recuperate homeostasis to the organelle [19]. The UPR^mt is induced upon impairments in mitochondrial function arising from sub-optimal mitochondrial protein folding [20,21], mitochondrial reactive oxygen species (ROS) production [22] or pathogenic infection [23]. This pathway induces a potent cell-survival response by promoting detoxification of ROS, mitochondrial protein homeostasis (by increasing the transcription of mitochondrial chaperones and proteases) and immune competence [20,23]. The UPR^mt is regulated by the basic leucine zipper (bZIP) transcription factor ATFS-1 in the model

organism and nematode *Caenorhabditis elegans* [20], with the bZIP transcription factor ATF5 from *Homo sapiens* (Hs-ATF5) mediating a mammalian UPR$^{mt}$ [24]. Regulation of the UPR$^{mt}$ occurs via organelle partitioning where ATFS-1/Hs-ATF5 are imported into healthy mitochondria and proteolytically degraded [20,24], which is dependent on the mitochondrial targeting sequence (MTS) of these proteins. Mitochondrial protein import efficiency is impaired in compromised mitochondria [25], allowing ATFS-1/Hs-ATF5 to localize to the nucleus through its nuclear localization signal whereupon it regulates a diverse set of genes promoting mitochondrial recovery [20,24].

As the UPR$^{mt}$ regulates many of the elements thought to be important in coral stress responses including production of heat shock proteins (HSP) and antioxidants, and that the regulatory pathways in coral remain largely obscure, we sought to characterize a possible UPR$^{mt}$ in the reef-building coral *Orbicella faveolata*. In this investigation, we demonstrate the existence of a pathway in *O. faveolata* which bears high similarity to the described UPR$^{mt}$. By using transgenesis of a genetic reporter line of UPR$^{mt}$ activity in *C. elegans*, we show that *O. faveolata* possesses a gene which is able to rescue a loss of function mutation of the UPR$^{mt}$ mediator ATFS-1 *in vivo*. We also demonstrate that due to its increased expression, Of-ATF5 may function in coral during both immune challenge and heat stress. In addition, by using bioinformatic methods, we determine that this transcription factor is associated with a mitochondrial-protective pathway that contains well-known stress resistance genes previously identified in reef-building corals [26–28]. Overall, our data suggest that the UPR$^{mt}$ could play a key role in mediating the ability of corals to adapt to a changing world.

## 2. Material and methods

### (a) UPR$^{mt}$ homology
Of-ATF5 was found with the tblastn algorithm available from NCBI, using *C. elegans* ATFS-1 or Hs-ATF5 as the query sequence with an *e*-value cutoff of $1 \times 10^{-5}$. Protein sequence alignments were performed through the TCOFFEE online alignment tool [29], and predictions of mitochondrial targeting sequences were performed with the online tool MITOPROT 2 [30]. Gene-tree analysis was performed in MEGA7 [31] by creating a consensus maximum-likelihood tree over 100 iterations. Species sequences: (*Acropora digitifera*, *Stylophora pistillata*, *Orbicella faveolata*, *Exaiptasia pallida*, *Danio rerio*, *Homo sapiens*, *Mus musculus*) were downloaded from NCBI. Protein sequences were found through the blastp algorithm using either Of-ATF5 or Hs-ATF5 for cnidarian or vertebrate species respectively.

### (b) Transgenesis
#### (i) Worm and bacterial strains
The reporter worm strains hsp-60$_{pr}$::GFP (SJ4058) and atfs-1(tm4525) hsp-60$_{pr}$::GFP used have been previously described [32,33]. Hermaphrodite worms were raised on the OP50 strain of *Escherichia coli* unless they were treated with RNAi, in which case the HT115 *E. coli* strain expressing the described RNAi plasmid was used. *C. elegans* strains were raised on nematode growth media plates (NGM) at either 16°C, 20°C or 25°C while the Of-ATF5 transgenic worms were maintained at 16°C unless stated otherwise.

#### (ii) Plasmid construction
Total RNA was isolated from adult polyps of *O. faveolata* using RNAqueous Total RNA Isolation kit (ThermoFisher scientific,

USA AM1912) according to the manufacturer's instructions. cDNA was obtained from total RNA using iScript cDNA synthesis kit (BioRad, USA 1708890) following the manufacturer's instructions. Of-ATF5 cDNA was amplified using primer pair Of-ATF5F (5′-TTTGGATCCATGGCCAGAACTTATCACAA-3′) and Of-ATF5R (5′-TTTGATATCTTATGAAGCAAGAAACACT-3′) and cloned into BamHI and EcoRV sites of the *C. elegans* expression vector pPD49.78, resulting in hsp-16$_{pr}$::Of-ATF5 [20]. The sequence of the cloned cDNA was confirmed by Sanger sequencing. The plasmid pPD49.78 includes the heat shock inducible promoter hsp-16.2 which we used to conditionally express Of-ATF5. Transgenic *C. elegans* was generated by co-injecting hsp-16$_{pr}$::Of-ATF5 (10 ng $\mu l^{-1}$) with a myo-2$_{pr}$::mCherry (5 ng $\mu l^{-1}$) marker plasmid and pBluescript (120 ng $\mu l^{-1}$) carrier plasmid into hsp-60$_{pr}$::GFP;atfs-1 (tm4525), generating extra-chromosomal arrays.

#### (iii) RNAi
RNAi was performed as previously described [34]. Briefly, worms were fed *E. coli* bacteria harbouring plasmids expressing double-stranded RNA for the mitochondrial quality control protease spg-7 or the ATP synthase subunit atp-2, both of which are capable of activating the UPR$^{mt}$ (figure 1e). Using qPCR, we calculated the percentage knockdown of atp-2 and spg-7 after RNAi to be 87% ($\pm$ 0.0009) and 77% ($\pm$0.007), respectively (electronic supplementary material, figure S2).

#### (iv) Microscopy
*Caenorhabditis elegans* were imaged using a Zeiss AxioCam MRm mounted on a Zeiss Imager Z2 microscope. Exposure times were the same in each experiment. Fluorescence was quantified using the program ImageJ [35] and the relative intensity between worm strains raised on each RNAi clone were compared using a one-way ANOVA using the tm4525 strain as a reference.

### (c) Bioinformatic analysis
#### (i) LPS experiment
Transcriptome analysis comes from a previous study where detailed methods can be found [36]. Briefly, 10 colonies of *O. faveolata* were collected near La Parguera, Puerto Rico. Samples were exposed to 1 ml of 7.57 mg ml$^{-1}$ lipopolysaccharide (LPS) from *E. coli* 0127:B8 (Sigma-Aldrich L3129-100MG) or vehicle control (filtered seawater) and incubated for 4 h upon which time all colonies were flash frozen. RNA was extracted with the RNAaqueous kit with DNAase step (Life Technologies AM1914) according to the manufacturer's instructions and quality assessed using the Agilent Bioanalyzer 2100. Samples with a RIN greater than 8 were used to create cDNA libraries with Illumina TruSeq RNA with Poly-A selection library kit. Libraries were sequenced by the University of Texas Southwestern Medical Center Genomics Core facility. Reads were filtered for adaptors and low-quality reads using TRIMMOMATIC software with default parameters [37] and mapped to an existing *O. faveolata* reference transcriptome [38] with the cufflinks package using default parameters [39], and read counts were generated by HTseq [40]. Read normalization was performed in the R package DESeq2 [41] to generate rlog transformed normalized reads. Of-ATF5 expression level from the rlog normalization was used in an unpaired *t*-test ($n = 4$ per group). The reference transcriptome was annotated using blastx algorithm against the Uniprot ensemble database with an *e*-value cut off of $1 \times 10^{-5}$. All raw sequences used in this project are available on the NCBI GenBank database (SRA Accession #SRP094633).

#### (ii) Weighted gene co-expression network analysis
Rlog-transformed counts were subject to weighted gene co-expression network analysis [42] which creates groups of

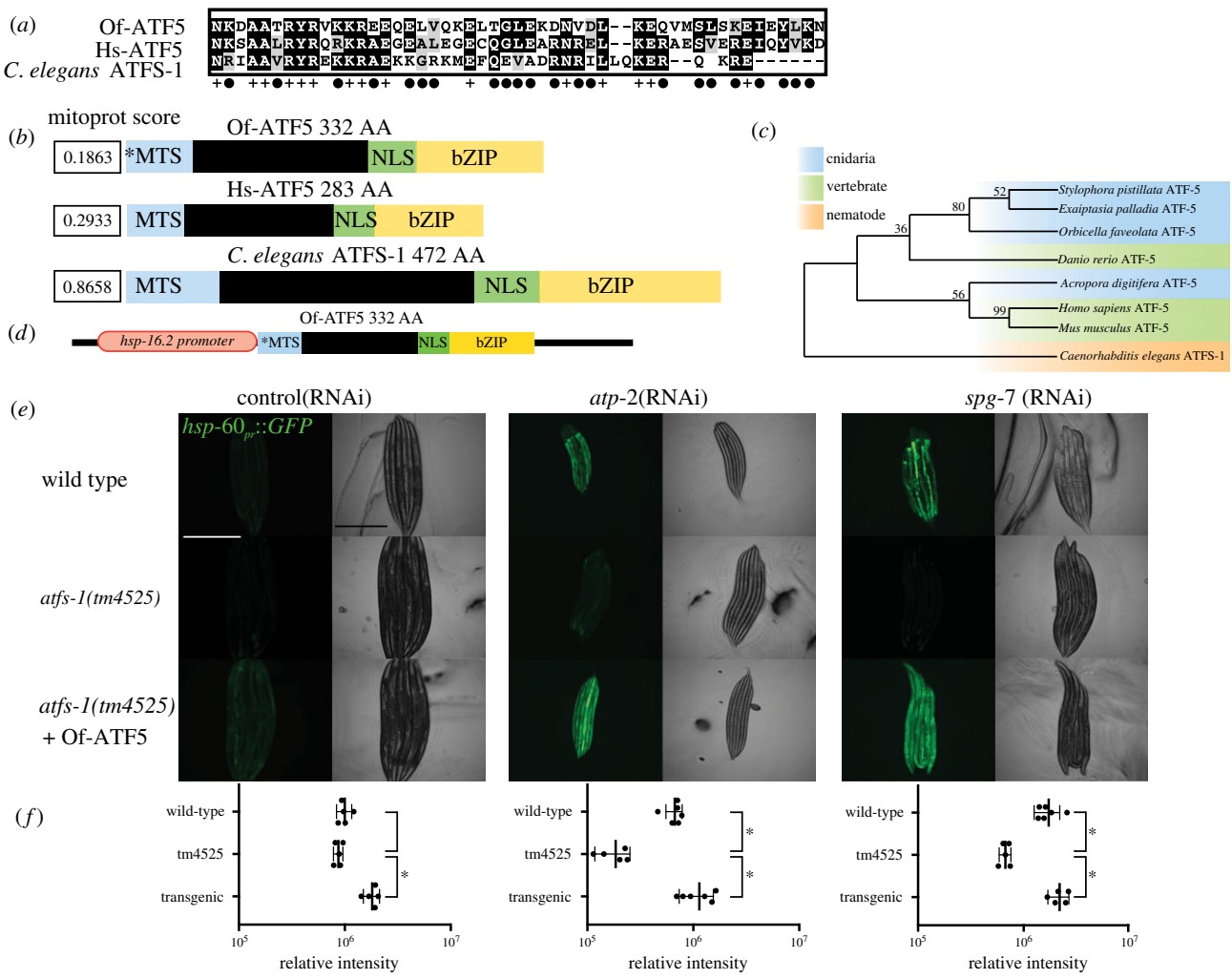

**Figure 1.** Expression of Of-ATF5 rescues UPR$^{mt}$ activity in worms lacking ATFS-1. (a) Alignment of amino acid sequence of bZIP domains in the homologous transcription factors *O. faveolata* (Of-ATF5), *Homo sapiens* (Hs-ATF5), *C. elegans* (ATFS-1). (+) represents consensus between all three species, (●) represents amino acid similarity between two species. (b) Schematic comparing the homologous bZIP transcription factors: Of-ATF5, Hs-ATF5, *C. elegans* ATFS-1. MTS-mitochondrial targeting sequence with Mitoprot scores (*MTS denotes unconfirmed MTS), NLS-nuclear localization sequence, bZIP-basic leucine zipper, AA-amino acid number. (c) Gene tree of ATF5/ATFS-1 homologues across multiple species demonstrating *C. elegans* ATFS-1 as outgroup to all other sequences, with numbers at nodes representing support of each association. (d) Transgene construct pPD49.78 expression plasmid with Of-ATF5 coding sequence insert downstream of the *C. elegans* hsp-16.2 promoter. (e) Photomicrograph of wild type (hsp-60pr::GFP), atfs-1(tm4525) expressing Of-ATF5 raised on control (HT115 RNAi), *atp*-2 or *spg*-7 RNAi. Scale bar, 0.5 mm. (f) Relative intensity of the fluorescent signal (from figure 1e) shown on a log scale (*p < 0.01). (Online version in colour.)

co-expressed contigs (modules), based on expression similarity [42]. Network construction was performed with an unsigned Pearson's correlation to generate modules with a power of 15, minimum module size of 30 and merge cut height of 0.25. The behaviour of the identified co-expression network modules was investigated with respect to three traits: treatment condition, genotype and expression of Of-ATF5.

### (iii) Transcription factor binding predictions
DNA-binding sequence preference provided as a positional weight matrix (PWM) for Hs-ATF5 (electronic supplementary material, figure S2) was obtained from the online resource CisBp [43,44] as the sequence preference of DNA-binding domains has a deep homology and is often extremely well conserved [43,45]. To investigate if the genes identified by our weighted gene co-expression network analysis (WGCNA) module possessed Hs-ATF5 binding motifs within their regulatory regions, we extracted 1000 bp upstream of the start codon for all annotated genes in the *O. faveolata* genome (ofav_dov_v1, GenBank: MZGG00000000.1) using a custom Python script and BEDTOOLS [46]. To identify motifs which match the PWM of Hs-ATF5 we used the program FIND INDIVIDUAL MOTIF OCCURRENCE (FIMO) [47]

to scan the regulatory region of the genes within the WGCNA module which was correlated to Of-ATF5. To investigate the enrichment of Hs-ATF5 binding motifs, we additionally scanned the regulatory region of a set of 'random genes', which were selected from a WGCNA module which had a minimal correlation to Of-ATF5 ($R = -0.147$, $p = 0.727$) and contained a large number of contigs (1676). To test for enrichment, the percentage of genes with motifs which matched the PWM of Hs-ATF5 between the two gene sets was compared using a Fisher's exact test.

### (iv) Gene ontology enrichment analysis
Genes which appeared in the highly correlated module and possessed annotations were used in the R script Gene Ontology enrichment analysis with Mann–Whitney U-test (GOMWU) [48]. Module membership scores acquired from WGCNA were used as a continuous trait for the genes in the highly correlated module, while all other genes in the transcriptome were given a significant measure of zero. Tests were performed to generate GO enrichment terms for the biological process with parameters: cluster cut height = 0.25, largest = 0.1, smallest = 25 and cellular compartment with parameters: cluster cut height = 0.25,

largest $= 0.1$, smallest $= 25$, with an option for modules analysis from WGCNA.

## (d) Temperature stress experiment

In June 2017, six colonies of *O. faveolata* were collected from Brewer's Bay St. Thomas under the Indigenous Species Research and Export Permit number CZM17010T, and split into two corresponding fragments with a diamond bladed saw and housed at the University of the Virgin Islands flow-through seawater facility to acclimate for two weeks. For the experiment, each coral fragment was placed into separate containers and either held at ambient temperature ($27.5°C$, STD $= 0.36$) or subjected to thermal stress ($29°C$, STD $= 0.844$), which reached $30°C$ over the course of 18 h, upon which time all colonies were flash frozen.

### (i) qPCR

RNA was extracted using Ambion RNeasy kit (ThermoFischer, USA: AM1920) and converted into cDNA using the iScript cDNA synthesis kit (BioRad, USA: 1708890). For qPCR reactions, 500 ng of cDNA was used in each well and samples were run in triplicate for each gene with Universal SYBR Green mix (BioRad, USA: 1725271). Target genes were selected due to known involvement in the UPR$^{mt}$ of *C. elegans*, heat shock protein 60 (HSP-60) and mitochondrial superoxide dismutase (mtSOD), the homologue of mammalian mitochondrial heat shock protein 70 (mtHSP-70), and translocase of inner mitochondrial membrane 23 (TIMM-23) [20]. Additionally, we pursued mitochondrial inner membrane protease 2 (IMP-2) as this gene was present in our highly significant WGCNA module. Expression was normalized to coral housekeeping gene eukaryotic initiation factor 3 (EIF3) [49] and fold-induction values were calculated with the $\Delta\Delta$ Ct method. Statistical analysis for Of-ATF5 expression was calculated with an unpaired *t*-test $n = 5$ per group. Correlation analysis of UPR$^{mt}$ genes was performed with Pearson's correlation between Of-ATF5 and target gene expression for each target gene separately. Primer design was accomplished through the use of Primer3 online tool [50] and primers are listed in electronic supplementary material, table S3.

## (e) Statistics

All statistical analysis including DESeq2, WGCNA and GOMWU was performed in the R programming environment [51].

## 3. Results

We were unable to locate an obvious ATFS-1 homologue in the genome of *O. faveolata* (electronic supplementary material, figure S1); however, a subsequent search revealed a putative homologue of Hs-ATF5 in the *O. faveolata* genome that contains a bZIP domain and a weakly predicted MTS termed Of-ATF5 (figure 1*a,b*). Both Of-ATF5 and Hs-ATF5 have weak MTS predictions, reflecting the resemblance between Hs-ATF5 and Of-ATF5 but not to ATFS-1. We therefore performed an additional analysis to investigate the presence of homologous ATF5/ATFS-1-like proteins across eight species: four symbiotic cnidarians, three vertebrate species and *C. elegans*. The created gene tree indicates that *C. elegans* ATFS-1 is an outgroup to all other sequences (figure 1*c*) reflecting our ability to locate an Hs-ATF5, but not ATFS-1, homologue in *O. faveolata*.

We conducted a transgenesis experiment to investigate if Of-ATF5 is functionally orthologous to ATFS-1 by determining if Of-ATF5 can functionally replace a loss-of-function *atfs-1* mutant in a UPR$^{mt}$ genetic reporter line of *C. elegans*. Here, we used the transgenic *C. elegans* strain

SJ4058 that contains a transcriptional green fluorescent protein (GFP) reporter for the mitochondrial chaperone gene *hsp-60* ($hsp-60_{pr}$::GFP), the promoter of which is directly bound by ATFS-1 during the UPR$^{mt}$ [20,32,52] (electronic supplementary material, figure S1). We used two different sources of mitochondrial stress to induce the expression of $hsp-60_{pr}$::GFP; RNAi knockdown of the mitochondrial quality control protease *spg-7* or the ATP synthase subunit *atp-2*. As expected, $hsp-60_{pr}$::GFP was induced in wild-type animals but not in the *atfs-1* loss-of-function mutant in the presence of these mitochondrial stress conditions (figure 1*e,f*). Conditional overexpression of Of-ATF5 in the *atfs-1* mutant background could restore $hsp-60_{pr}$::GFP expression in the presence of mitochondrial stress (figure 1*e,f*), suggesting that Of-ATF5 may constitute a bona fide homologue of ATFS-1/Hs-ATF5.

## (a) Induction of a UPR$^{mt}$ during immune stress

Of-ATF5 expression is increased during an immune challenge (figure 2*a*) ($p < 0.0001$), consistent with expectations. To computationally identify networks of genes [53] associated with Of-ATF5 expression, we employed WGCNA [42]. We identified one WGCNA module which was significantly correlated to the expression level of Of-ATF5 ($R = 0.94$, $p < 0.001$) containing 941 contigs with gene annotations (figure 2*b*; electronic supplementary material, table S1 and figure S2), of which 91 are mitochondrially localized (electronic supplementary material, table S1). Of-ATF5 is present as one of the core module genes ($p < 0.001$) (electronic supplementary material, table S1) indicating that this transcriptional network is robustly connected to Of-ATF5. Of the 941 genes identified by our WGCNA analysis, we were able to identify and extract the regulatory region of 807 (86%) from the *O. faveolata* genome. By using transcription factor binding predictions, we identified Hs-ATF5 binding motifs within the regulatory region of 181 of these genes (22.4%) (electronic supplementary material, table S2), which is a significant enrichment over the number of Hs-ATF5 binding motifs within the regulatory region of 'random' genes (3.32%, $p < 2.2 \times 10^{-16}$, Fisher's exact test). Gene Ontology analysis of our significantly correlated WGCNA module revealed enrichment of several mitochondrial cellular compartment terms (figure 2*c*) and biological processes including mitochondrial transmembrane transport, proteolysis, regulation of cell death and oxidation-reduction process (figure 2*d*).

## (b) Induction of the coral UPR$^{mt}$ during temperature stress

Elevated temperatures are able to activate the two UPR$^{mt}$ reporter lines $hsp-60_{pr}$::GFP and $hsp-6_{pr}$::GFP in *C. elegans* (electronic supplementary material, figure S1). We therefore tested if Of-ATF5 expression is increased in colonies of *O. faveolata* during pre-bleaching thermal stress of $2°C$ above ambient and found it to be increased relative to the control (figure 3*a*) ($p < 0.05$). The expression of our five investigated mitochondrial-protective target genes had a positively correlated expression with Of-ATF5 (figure 3*b–f*) ($p < 0.05$) consistent with the expected relationship between a transcription factor and a target gene [54]. For one of the samples, we were unable to obtain amplification for either our mtSOD or TIMM-23 primers and for these genes only four of our five samples were used for this calculation.

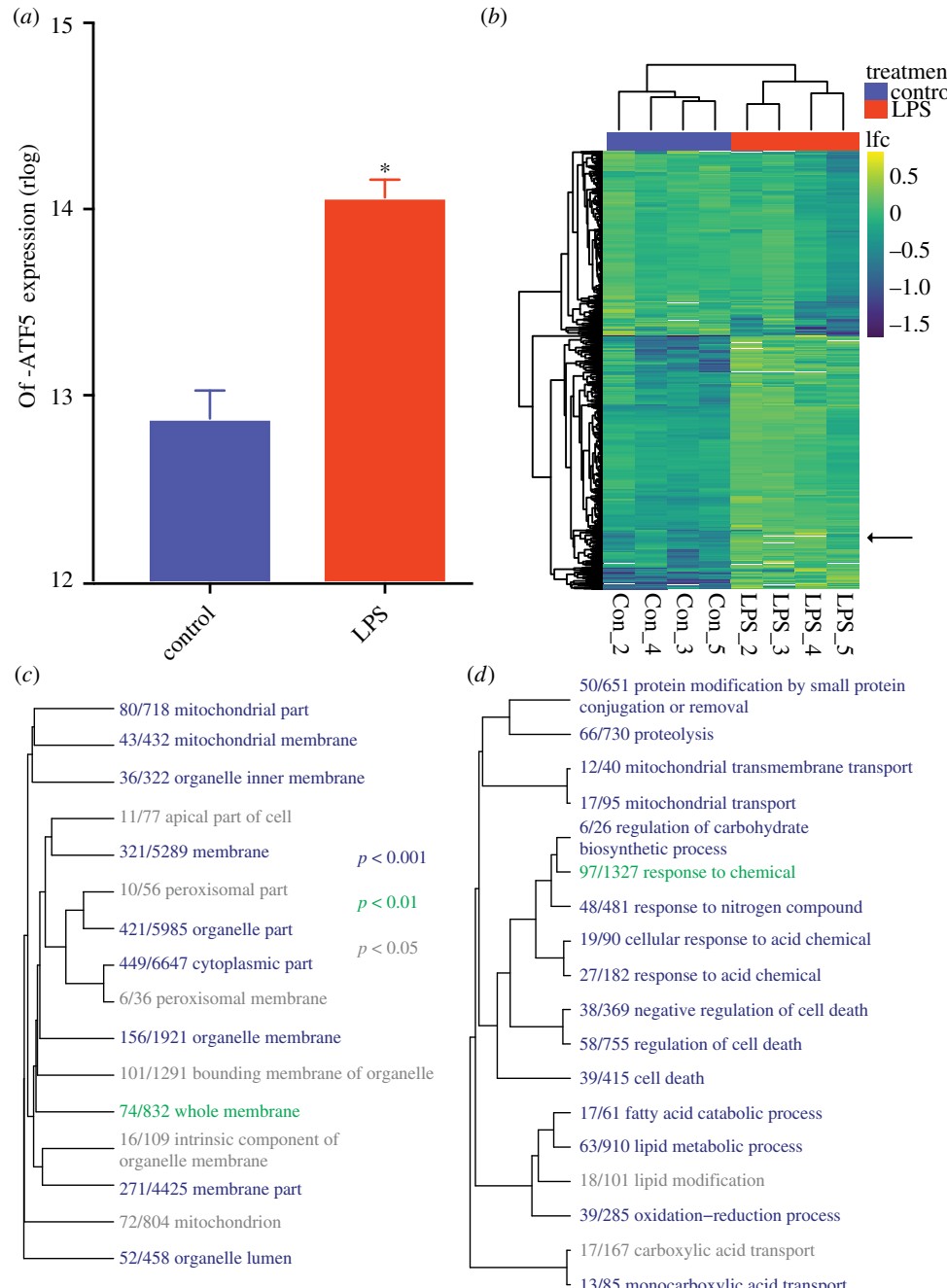

**Figure 2.** Induction of a coral UPR$^{mt}$ during immune challenge. (a) Expression of Of-ATF5 during control or immune challenge (LPS). $p < 0.0001$; bar represents s.d.; $n = 4$ per group. (b) Heatmap depicting the log$_2$ fold change of annotated contigs in the highly correlated WGCNA module during LPS treatment. Of-ATF5 is denoted with an arrow. (c,d) Gene ontology category enrichment analysis. Dendrograms depict the sharing of genes between categories; colours indicate significance of each term (Mann–Whitney $U$ test) as indicated by the inserted key. Fraction indicates number of genes assigned to a specific GO term contained in the WGCNA module over the total number of genes possessing that GO term in the transcriptome for (c) cellular compartment and (d) biological process. (Online version in colour.)

## 4. Discussion

### (a) Uncovering a UPR$^{mt}$ in *O. faveolata*

Identifying conserved cellular pathways such as the UPR$^{mt}$ in environmentally important basal metazoans like coral has implications in understanding both the evolutionary roots of stress-response pathways as well as molecular mechanisms promoting adaptability to a rapidly changing environment. By using multiple complementary approaches (transgenesis, bioinformatics and qPCR), we uncovered members of a putative UPR$^{mt}$ pathway in *O. faveolata* which is induced during both immune challenge and temperature stress. Overall, our data demonstrate that the UPR$^{mt}$ appears to be a well-conserved stress-response pathway [55] at the base of animal

evolution, which probably has significant implications in a coral's capacity to respond to environmental stressors.

We were able to establish functional conservation of Of-ATF5 as it can rescue UPR$^{mt}$ activity in a *C. elegans atfs-1* loss-of-function mutant, demonstrating *in vivo* functional replacement using a coral gene. Furthermore, because Of-ATF5 could only restore UPR$^{mt}$ activity under conditions that perturb mitochondrial function, it suggests that Of-ATF5 may be regulated by mitochondrial import efficiency in a manner akin to ATFS-1 or Hs-ATF5 [20,24].

The promising results of our transgenesis experiment led us to investigate if Of-ATF5 mediates a UPR$^{mt}$-like pathway in *O. faveolata*. We conclude that Of-ATF5 likely mediates a UPR$^{mt}$ in *O. faveolata* which is similar to the response

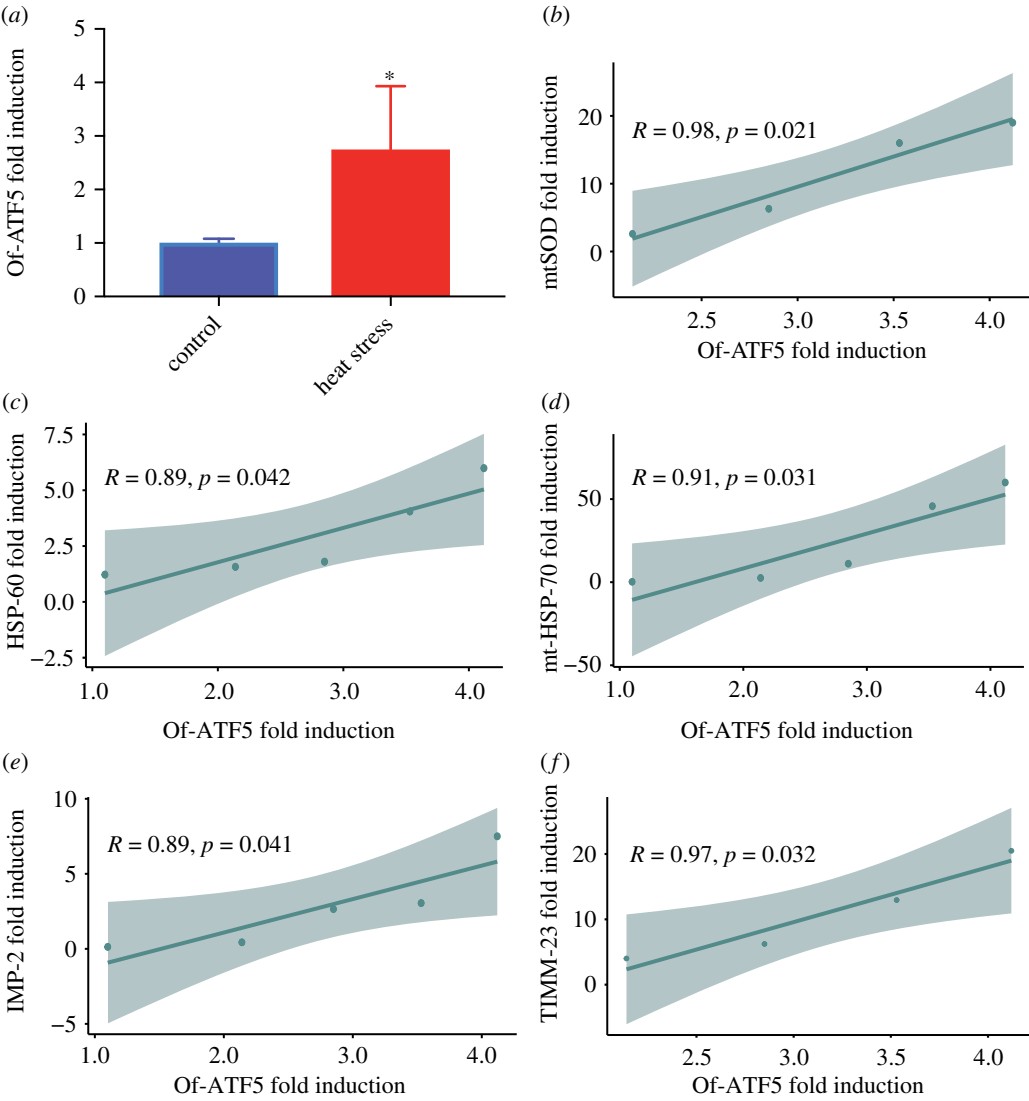

**Figure 3.** Induction of the UPR$^{mt}$ during heat stress. (*a*) Expression of Of-ATF5 during control or heat stress. $p < 0.05$; bar represents s.d.; $n = 5$ per group. (*b–f*) Correlation between Of-ATF5 and mitochondrial superoxide dismutase (mtSOD) (*b*), heat shock protein 60 (HSP-60) (*c*), mitochondrial heat shock protein 70 (mtHSP-70) (*d*), mitochondrial inner membrane protease-2 (IMP2) (*e*) and translocase of inner mitochondrial membrane-23 (TIMM-23) (*f*). Correlation coefficient (*R*) and *p*-values are shown on in-graph insert, dots represent individual samples and shaded area represents 95% confidence interval. (Online version in colour.)

mediated by ATFS-1 in *C. elegans*, by regulating the expression of genes that are involved in protein homeostasis [56] and the detoxification of damaging free radicals [20]. Furthermore, the UPR$^{mt}$ probably has significant implications for a coral's capacity to respond to environmental conditions that perturb mitochondrial function, since it is induced during two conditions affecting corals on a global scale, immune challenge and temperature stress.

## (b) The role of the UPR$^{mt}$ in coral disease

Marine diseases are changing the face of reefs worldwide [3,4,6,57], and the mechanisms which corals use to overcome pathogens remain poorly characterized. The UPR$^{mt}$ of *C. elegans* is induced during bacterial infection and serves to promote immune competence by improving both pathogen clearance and tolerance during infection [23,58]. We found support that Of-ATF5 likewise is upregulated during the immune challenge with LPS [36], an endotoxin found in the outer membrane of bacteria that is a potent inducer of the immune response in both invertebrate and vertebrate model systems. Using gene network analysis and transcription factor binding site predictions, we found support that

Of-ATF5 possibly functions to directly regulate a broad transcriptional stress response during the immune challenge. Based upon the high level of similarity between the responses mediated by Of-ATF5 and ATFS-1, we generated a hypothetical model where the UPR$^{mt}$ of *O. faveolata* might function similarly to the UPR$^{mt}$ of *C. elegans* by promoting mitochondrial recovery during immune challenge (figure 4). Mitochondria are purveyors of innate immunity and mediators of cell death [63,64], and future investigations should explore if the immune promoting abilities of the UPR$^{mt}$ are conserved in corals.

## (c) The role of the UPR$^{mt}$ in a warming climate

Coral bleaching induced by elevated temperature disrupts the physiology of coral-dinoflagellate symbiosis [65,66], which involves excessive ROS production [10]. Our findings support previous work by Voolstra *et al.* [26], who identified Of-ATF5 as part of the response to elevated temperature in *O. faveolata*. The UPR$^{mt}$ functions to concomitantly promote the detoxification of ROS while simultaneously minimizing its production [20,52]. Interestingly, of the mitochondrial genes which had correlated expression with Of-ATF5 during temperature stress, both

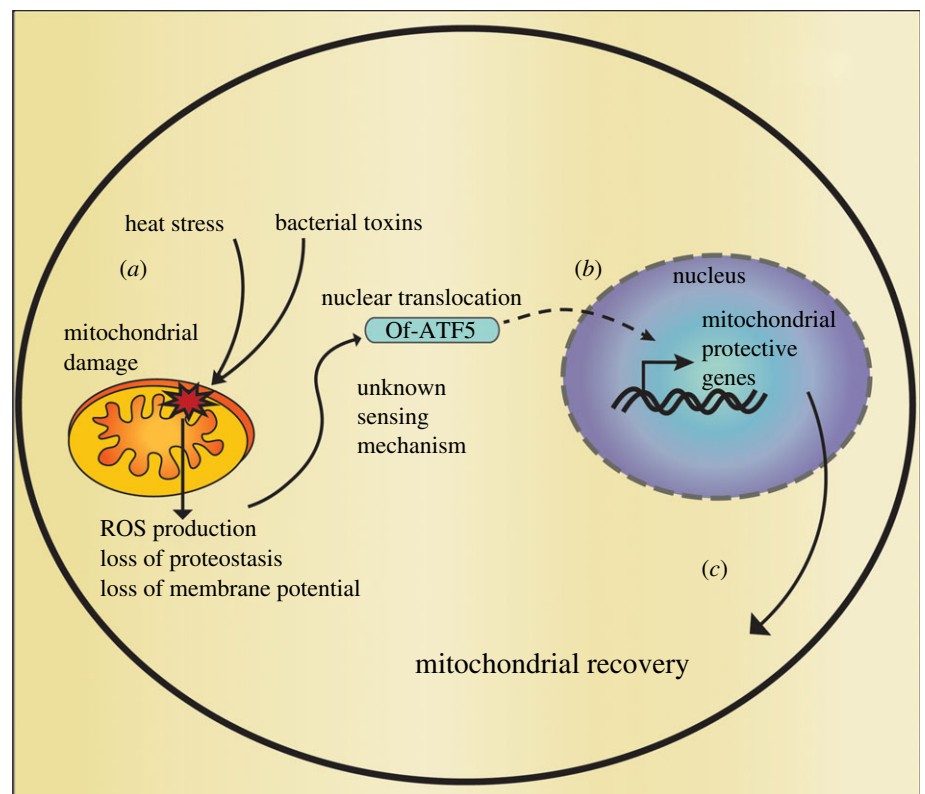

**Figure 4.** Hypothetical model of UPR$^{mt}$ induction during periods of environmental stress in *O. faveolata*. (*a*) Mitochondrial damage occurs through the effects of LPS and/or other mitochondrial targeted bacterial toxins [17,59,60], or elevated temperatures [18]. Mitochondrial damage results in increased production of ROS [61], loss of mitochondrial proteostasis and membrane potential [62]. (*b*) Mitochondrial damage is sensed through an unknown mechanism, which causes the increased activity of Of-ATF5 and subsequent UPR$^{mt}$ induction. (*c*) Upon activation of the UPR$^{mt}$, the expression of mitochondrial-protective genes is increased, which probably leads to mitochondrial recovery. (Online version in colour.)

mtHSP-70 and TIMM-23 are known genes involved in the *C. elegans* UPR$^{mt}$ [20] and were also identified by our WGCNA analysis. Thermal stress studies in symbiotic cnidarians have demonstrated dysfunctions in cellular processes consistent with target genes that form part of the UPR$^{mt}$, including protein misfolding [67,68], ROS production and mitochondrial damage [18,69,70]. We therefore developed a hypothetical model of a mitochondrial stress-based mechanism of dysfunction (figure 4) as a parsimonious explanation for the involvement of the mitochondria in contributing to the physiology of cnidarians during temperature anomalies. In support of our model, increased expression of mitochondrial chaperones and antioxidants which are likely to be mediated by the UPR$^{mt}$ have been associated with improved thermal tolerance in corals [14,16,28,71] indicating that the UPR$^{mt}$ may be a contributing pathway mediating corals ability to adapt to a warming ocean.

## 5. Conclusion

Despite worldwide declines, the cellular mechanisms which promote the ability of corals to survive and adapt to a changing ocean are not yet fully characterized. If these pathways can be uncovered then active interventions to restore and protect the world's reefs, including coral restoration, assisted evolution and or gene flow [72,73], can become more effective. In lieu of our data, Of-ATF5 may show promise as a target gene for further investigation as it mediates a mitochondrial-protective gene network during exposure to two prominent environmental stressors: disease and hyperthermic-temperature. Overall our results point towards the potential of the UPR$^{mt}$ to allow reef-building corals to persist under mounting environmental stress in a rapidly changing ocean.

Data accessibility. Raw reads from this study are available on the NCBI Sequence Read Archive (SRA Accession #SRP094633), and all codes and data used in this study are available on the Dryad Digital Repository: https://doi.org/10.5061/dryad.40sf920 [74].

Authors' contributions. Conceptualization, B.A.D., M.W.P., L.D.M.; genetic method expertise, S.A.M., M.W.P.; bioinformatic expertise, L.E.F., data analysis., B.A.D, S.A.M.; visualization, B.A.D; writing, B.A.D., L.D.M., M.W.P.; funding acquisition, L.D.M., M.W.P.

Competing interests. The authors declare no competing interests.

Funding. This work was supported by NSF Biological Oceanography Award Number 1712134 awarded to L.D.M. and University of Texas at Arlington Startup grant awarded to M.W.P.

Acknowledgements. We would like to thank Marylin Brandt for collection of *O. faveolata*, Contessa Ricci and Nicholas Macknight for carrying out the temperature stress experiment.

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
