## [Reviewer comments · Proceedings of the Royal Society B: Biological Sciences]

Review History

RSPB-2019-0470.R0 (Original submission)

Review form: Reviewer 1

Recommendation

Accept with minor revision (please list in comments)

Scientific importance: Is the manuscript an original and important contribution to its field?

Excellent

General interest: Is the paper of sufficient general interest?

Excellent

Quality of the paper: Is the overall quality of the paper suitable?

Excellent

Is the length of the paper justified?

Yes

Should the paper be seen by a specialist statistical reviewer?

No

Do you have any concerns about statistical analyses in this paper? If so, please specify them explicitly in your report.

Yes

It is a condition of publication that authors make their supporting data, code and materials available - either as supplementary material or hosted in an external repository. Please rate, if applicable, the supporting data on the following criteria.

Is it accessible?

Yes

Is it clear?

Yes

Is it adequate?

Yes

Do you have any ethical concerns with this paper?

No

Comments to the Author

This is a truly trailblazing study, the first one to go all the way to validating the presumed molecular mechanism in corals. It is crystal clear, well written, and combines multiple lines of evidence including rescue of the loss of function mutation in *C.elegans* by a transgenically expressed coral gene.

Major concerns:

Search for transcription binding sites for ATF5 was limited to “likely” genes. Would a search among “unlikely” or just randomly picked genes be significantly less successful? Please demonstrate that, otherwise the argument is incomplete.

Unsigned WGCNA was performed, meaning that modules would include both positively and negatively correlated genes (as is clear from Fig. 2B). Is this choice justified for ATF5? It would be a reasonable choice if ATF5 could both induce and repress expression of target genes; but since it is mainly inducing (as seems to be the case), signed WGCNA analysis seems more biologically appropriate. Did you try that? (that said, finding of a highly ATF5-correlated WGCNA module with strong enrichment for mitochondrial processes is extremely neat!)

Fig 3B contains only 4 points, and 3F - only 3 points of the original 5 (missing one or two less-induced points). Why? Please do show them if you have data.

Does Fig 3 show ALL the genes that were tested for correlation with ATF5?

L231-235: Phylogenetic analysis of ATF5 is mentioned here for the first time. Please add this to the Results - and include a figure!

L255: “UPRmt of *O. faveolata* might function similarly to the UPRmt of *C. elegans* by promoting immune competence (figure 4). Overall, this indicates that the role mitochondria as purveyors of innate immunity (57, 58) and mediators of cell death has a deep evolutionary origin and functions

in the immunology of corals” – there is no notion of immune competence on Fig.4, and, as far as I can see, there is no data here to suggest that function Of-ATF5 affects coral immunity (although it does respond to the immune challenge).

(on a similar note) L237-240: “We conclude that Of-ATF5 likely mediates a UPRmt in *O. faveolata* which is similar to the response mediated by ATF5-1 in *C. elegans*, by regulating the expression of genes that are involved in protein homeostasis (54), detoxification of damaging free radicals (20), and innate immunity (23).” – technically, what is demonstrated here is that it Of-ATF5 can regulate a heat shock protein in *C.elegans*, is correlated with mitochondrial genes and genes involved in cell death (by GO analysis in a coral), and with an antioxidant gene mtSOD; but there seems to be no direct evidence for its involvement in immunity.

To be honest I find the whole discussion section about “The role of the UPRmt in coral disease” premature. The paper nicely establishes that corals possess UPRmt response, but it remains to be seen what role does it actually play in disease and immunity. It responds to immune challenge (which is nicely shown), but this could be a response to general cellular stress rather than a specific immunity mechanism (for example, under any stress ribosomal genes tend to get down-regulated, reflecting decrease in growth).

Minor things:

Figure 3 legend – please spell out the names of proteins correlated with ATF5 here.

Figure 4 is a general summary of ATF5 function rather than a conclusion of this study (if we just remove “Of-” from the label of the protein), so maybe make it Fig 1 and refer to it in intro?

Fig. 1A: I am not clear why Of-ATF5 has a long gap near the N-terminus after position 9. The alignment would be perfectly fine without this gap.

L304: Fig 2 legend: “Fraction indicates number of genes with values exceeding significance threshold in each term” – if the GO_MWU was run correctly for WGCNA modules, this fraction should be (number of genes from this GO category assigned to the module) / (total number of genes in this GO category). Please check!

Fig. 2B: is in this heatmap, please indicate with an arrow the row corresponding to Of-ATF5. (several places in text) “obscured” obscure

L122: “Rlog transformed contigs” Rlog transformed counts

L125-127: The last sentence in the WGCNA description creates an impression that the three named traits were used to guide the coexpression network construction, which is not how it works. I suggest rephrasing: “The behavior of the identified coexpression network modules was investigated with respect to three traits: ...”.

L133: “prompter” promoter

L155: “either held at either” either held at

L198: “figure 3B” - I am guessing it should be Fig. 2B.

L268: “We therefore developed model of a mitochondrial stress-based mechanism of dysfunction (figure 4)” – as I mentioned previously, this figure is in fact the summary of prior knowledge about Hs-ATF5 function in which “Hs” is substituted for “Of”, not the “model” substantiated by

the results of this study. For example, this work presents no direct evidence of nuclear translocation, of the involvement of ROS, loss of proteostasis, or loss of membrane potential in affecting Of-ATF5 function. There is also no evidence that ATF5 function actually leads to mitochondrial recovery.

Review form: Reviewer 2 (Virginia Weis)

Recommendation

Major revision is needed (please make suggestions in comments)

Scientific importance: Is the manuscript an original and important contribution to its field?

Good

General interest: Is the paper of sufficient general interest?

Acceptable

Quality of the paper: Is the overall quality of the paper suitable?

Good

Is the length of the paper justified?

Yes

Should the paper be seen by a specialist statistical reviewer?

Yes

Do you have any concerns about statistical analyses in this paper? If so, please specify them explicitly in your report.

No

It is a condition of publication that authors make their supporting data, code and materials available - either as supplementary material or hosted in an external repository. Please rate, if applicable, the supporting data on the following criteria.

Is it accessible?

Yes

Is it clear?

Yes

Is it adequate?

Yes

Do you have any ethical concerns with this paper?

Yes

Comments to the Author

This manuscript describes a mitochondrial unfolded protein response in corals by describing a transcription factor ATF5-like homolog in corals, examining its function in a transgenic system in *c. elegans*, and performing a bioinformatics analysis linking this gene with expression of other genes during an immune challenge and temperature stress response.

Overall I feel that the work was capably carried out. I appreciate the collaboration between coral biologists and geneticists from other fields who are bringing new perspectives and techniques to the field of coral biology. I do have some concerns that I feel can be easily addressed in a revision.

First and most importantly, I feel that the authors overplay their hand, especially in the rationale for carrying out the work. At this point, there are several dozen reviews on coral bleaching, indeed 3 within the past year alone. Although we do not have every detail on the mechanisms of bleaching, by any stretch, we do know quite a bit at this point. So wording in lines 32-35 and lines 259-283 should be modified – sure we don't know the level of detail on par with well-developed model systems, we nonetheless know a lot. The UPR is an interesting addition to the repertoire of bleaching mechanisms but it is not particularly surprising or unexpected. It slots right in with the variety of other stress/immune/ROS responses that have been studied and described for years. I feel that the authors need to tone down both the rationale and the conclusions section of the paper and provide the proper context for where UPR fits in the story.

There are areas in the manuscript where I do not feel that the coral biologists have added enough of their content knowledge. This is especially notable regarding the discussion of ATF5 homologies and trees (Fig S1 and lines 229-231). It is now well recognized that the Ecdysozoa are outgroups in the phylogenomics of the Metazoa, see for example Miller et al Genome Biology 2007. Genes and genomes of early diverging metazoans more closely resemble those Deuterostomes including chordates than the Ecdysozoans. So it is not at all surprising that coral ATF5 is closer to vertebrates than *C. elegans*. This should be modified in the discussion.

While I understand that length limits sometimes constrain explanations of methods and results, I feel in places that there is an absence of information. Figure 1D is a nice qualitative measure of knockdown and rescue, but where are quantitative data backing these findings up? The methods for Figure 2A are missing. What is rlog? QPCR? Other method? What statistics were performed? Explain. I have similar comment for the LPS experiment, lines 105-108. Could someone repeat this experiment using this description? How did you arrive at this concentration of LPS and the 4 hour incubation time. A supplementary figure and methods could explain pilot experiments that helped you arrive at this end point. What is the vehicle and at what concentration?

Review form: Reviewer 3

Recommendation

Major revision is needed (please make suggestions in comments)

Scientific importance: Is the manuscript an original and important contribution to its field?

Good

General interest: Is the paper of sufficient general interest?

Good

Quality of the paper: Is the overall quality of the paper suitable?

Acceptable

Is the length of the paper justified?

Yes

Should the paper be seen by a specialist statistical reviewer?

No

Do you have any concerns about statistical analyses in this paper? If so, please specify them explicitly in your report.

Yes

It is a condition of publication that authors make their supporting data, code and materials available - either as supplementary material or hosted in an external repository. Please rate, if applicable, the supporting data on the following criteria.

Is it accessible?

Yes

Is it clear?

Yes

Is it adequate?

Yes

Do you have any ethical concerns with this paper?

No

Comments to the Author

The manuscript's main ideas and general focus are sound. Broadly, it has been a long-standing goal in molecular evolution to reconstruct the evolutionary progression of major signaling pathways, including the unfolded protein response. Specifically, it has been a major focus in the field of cnidarian-Symbiodinium symbiosis to understand the genes and pathways involved in this symbiosis and its ultimate break down upon stress. The researchers take a strong molecular approach in both coral and a model system to understand the evolution of the mitochondria unfolded protein response. In general, coral reef research would be advanced by more studies like this one.

Towards this end, the authors provide some compelling evidence for a conserved response of an ortholog of ATF5 during both thermal and immune stress. In addition, the authors provide evidence that the ATF5-downstream response may be similar to what occurs in bilateral metazoans. It is my general feeling that the data presented suggest that ATF5 might function similarly in coral as other animals. However, the data do not warrant many of the paper's strong conclusions regarding the overall conservation of the UPRmt in corals and its putative role in protecting against bleaching. This pitfall can be improved through a combination of softening the conclusions along with some additional computational experiments as discussed below. (See Appendix A)

Decision letter (RSPB-2019-0470.R0)

04-Apr-2019

Dear Mr Dimos:

Your manuscript has now been peer reviewed and the reviews have been assessed by an Associate Editor. The reviewers' comments (not including confidential comments to the Editor) and the comments from the Associate Editor are included at the end of this email for your

reference. As you will see, the reviewers and the Editors have raised some concerns with your manuscript and we would like to invite you to revise your manuscript to address them.

Research ethics:

Use of animals and field studies:

Please submit a copy of your revised paper within three weeks. If we do not hear from you within this time your manuscript will be rejected. If you are unable to meet this deadline please let us know as soon as possible, as we may be able to grant a short extension.

Best wishes,

Proceedings B
mailto: proceedingsb@royalsociety.org

Associate Editor
Comments to Author:
Dear Bradford Dimos

Your manuscript has been assessed by three reviewers who all agree that your manuscript is interesting and sound, and that it could ultimately be accepted in Proc Royal Soc. They provide a number of general and specific suggestions around the points that would need to be improved before this can occur. In particular, they think that you overstate the importance of your results and neglect foundational knowledge in both the coral bleaching field and that of ortholog replacement studies. This shortcoming can be addressed by a careful revision of the text. Suggestions are also made on how to improve the statistical and bioinformatic analyses of the results and required information to be presented in the text to substantiate conclusions made.

Based on these reviews I would be happy to consider a revised version of your manuscript for further consideration in Proc Royal Soc. If you choose to revise and resubmit, please provide a

point-by-point response to all the reviewers comments and suggestions. In this response please refer to the line numbers in the track-change or final version of the ms where changes have been made.

Warm Regards
Line K Bay

Reviewer(s)' Comments to Author:

Referee: 1

Comments to the Author(s)

This is a truly trailblazing study, the first one to go all the way to validating the presumed molecular mechanism in corals. It is crystal clear, well written, and combines multiple lines of evidence including rescue of the loss of function mutation in *C.elegans* by a transgenically expressed coral gene.

Major concerns:

Search for transcription binding sites for ATF5 was limited to "likely" genes. Would a search among "unlikely" or just randomly picked genes be significantly less successful? Please demonstrate that, otherwise the argument is incomplete.

Unsigned WGCNA was performed, meaning that modules would include both positively and negatively correlated genes (as is clear from Fig. 2B). Is this choice justified for ATF5? It would be a reasonable choice if ATF5 could both induce and repress expression of target genes; but since it is mainly inducing (as seems to be the case), signed WGCNA analysis seems more biologically appropriate. Did you try that? (that said, finding of a highly ATF5-correlated WGCNA module with strong enrichment for mitochondrial processes is extremely neat!)

Fig 3B contains only 4 points, and 3F - only 3 points of the original 5 (missing one or two less-induced points). Why? Please do show them if you have data.

Does Fig 3 show ALL the genes that were tested for correlation with ATF5?

L231-235: Phylogenetic analysis of ATF5 is mentioned here for the first time. Please add this to the Results - and include a figure!

L255: "UPRmt of *O. faveolata* might function similarly to the UPRmt of *C. elegans* by promoting immune competence (figure 4). Overall, this indicates that the role mitochondria as purveyors of innate immunity (57, 58) and mediators of cell death has a deep evolutionary origin and functions in the immunology of corals" - there is no notion of immune competence on Fig.4, and, as far as I can see, there is no data here to suggest that function Of-ATF5 affects coral immunity (although it does respond to the immune challenge).

(on a similar note) L237-240: "We conclude that Of-ATF5 likely mediates a UPRmt in *O. faveolata* which is similar to the response mediated by ATF5-1 in *C. elegans*, by regulating the expression of genes that are involved in protein homeostasis (54), detoxification of damaging free radicals (20), and innate immunity (23)." - technically, what is demonstrated here is that it Of-ATF5 can regulate a heat shock protein in *C.elegans*, is correlated with mitochondrial genes and genes involved in cell death (by GO analysis in a coral), and with an antioxidant gene mtSOD; but there seems to be no direct evidence for its involvement in immunity.

To be honest I find the whole discussion section about “The role of the UPRmt in coral disease” premature. The paper nicely establishes that corals possess UPRmt response, but it remains to be seen what role does it actually play in disease and immunity. It responds to immune challenge (which is nicely shown), but this could be a response to general cellular stress rather than a specific immunity mechanism (for example, under any stress ribosomal genes tend to get down-regulated, reflecting decrease in growth).

Minor things:

Figure 3 legend – please spell out the names of proteins correlated with ATF5 here.

Figure 4 is a general summary of ATF5 function rather than a conclusion of this study (if we just remove “Of-” from the label of the protein), so maybe make it Fig 1 and refer to it in intro?

Fig. 1A: I am not clear why Of-ATF5 has a long gap near the N-terminus after position 9. The alignment would be perfectly fine without this gap.

L304: Fig 2 legend: “Fraction indicates number of genes with values exceeding significance threshold in each term” – if the GO_MWU was run correctly for WGCNA modules, this fraction should be (number of genes from this GO category assigned to the module) / (total number of genes in this GO category). Please check!

Fig. 2B: is in this heatmap, please indicate with an arrow the row corresponding to Of-ATF5.

(several places in text) “obscured” obscure

L122: “Rlog transformed contigs” Rlog transformed counts

L125-127: The last sentence in the WGCNA description creates an impression that the three named traits were used to guide the coexpression network construction, which is not how it works. I suggest rephrasing: “The behavior of the identified coexpression network modules was investigated with respect to three traits: ...”.

L133: “prompter” promoter

L155: “either held at either” either held at

L198: “figure 3B” - I am guessing it should be Fig. 2B.

L268: “We therefore developed model of a mitochondrial stress-based mechanism of dysfunction (figure 4)” – as I mentioned previously, this figure is in fact the summary of prior knowledge about Hs-ATF5 function in which “Hs” is substituted for “Of”, not the “model” substantiated by the results of this study. For example, this work presents no direct evidence of nuclear translocation, of the involvement of ROS, loss of proteostasis, or loss of membrane potential in affecting Of-ATF5 function. There is also no evidence that ATF5 function actually leads to mitochondrial recovery.

Referee: 2

Comments to the Author(s)

This manuscript describes a mitochondrial unfolded protein response in corals by describing an transcription factor ATF5-like homolog in corals, examining its function in a transgenic system in

c. *elegans*, and performing a bioinformatics analysis linking this gene with expression of other genes during an immune challenge and temperature stress response.

Overall I feel that the work was capably carried out. I appreciate the collaboration between coral biologists and geneticists from other fields who are bringing new perspectives and techniques to the field of coral biology. I do have some concerns that I feel can be easily addressed in a revision.

First and most importantly, I feel that the authors overplay their hand, especially in the rationale for carrying out the work. At this point, there are several dozen reviews on coral bleaching, indeed 3 within the past year alone. Although we do not have every detail on the mechanisms of bleaching, by any stretch, we do know quite a bit at this point. So wording in lines 32-35 and lines 259-283 should be modified – sure we don't know the level of detail on par with well-developed model systems, we nonetheless know a lot. The UPR is an interesting addition to the repertoire of bleaching mechanisms but it is not particularly surprising or unexpected. It slots right in with the variety of other stress/immune/ROS responses that have been studied and described for years. I feel that the authors need to tone down both the rationale and the conclusions section of the paper and provide the proper context for where UPR fits in the story.

There are areas in the manuscript where I do not feel that the coral biologists have added enough of their content knowledge. This is especially notable regarding the discussion of ATF5 homologies and trees (Fig S1 and lines 229-231). It is now well recognized that the Ecdysozoa are outgroups in the phylogenomics of the Metazoa, see for example Miller et al Genome Biology 2007. Genes and genomes of early diverging metazoans more closely resemble those Deuterostomes including chordates than the Ecdysozoans. So it is not at all surprising that coral ATF5 is closer to vertebrates than *C. elegans*. This should be modified in the discussion.

While I understand that length limits sometimes constrain explanations of methods and results, I feel in places that there is an absence of information. Figure 1D is a nice qualitative measure of knockdown and rescue, but where are quantitative data backing these findings up? The methods for Figure 2A are missing. What is rlog? QPCR? Other method? What statistics were performed? Explain. I have similar comment for the LPS experiment, lines 105-108. Could someone repeat this experiment using this description? How did you arrive at this concentration of LPS and the 4 hour incubation time. A supplementary figure and methods could explain pilot experiments that helped you arrive at this end point. What is the vehicle and at what concentration?

Referee: 3

Comments to the Author(s)

The manuscript's main ideas and general focus are sound. Broadly, it has been a long-standing goal in molecular evolution to reconstruct the evolutionary progression of major signaling pathways, including the unfolded protein response. Specifically, it has been a major focus in the field of cnidarian-Symbiodinium symbiosis to understand the genes and pathways involved in this symbiosis and its ultimate break down upon stress. The researchers take a strong molecular approach in both coral and a model system to understand the evolution of the mitochondria unfolded protein response. In general, coral reef research would be advanced by more studies like this one.

Towards this end, the authors provide some compelling evidence for a conserved response of an ortholog of ATF5 during both thermal and immune stress. In addition, the authors provide evidence that the ATF5-downstream response may be similar to what occurs in bilateral metazoans. It is my general feeling that the data presented suggest that ATF5 might function

similarly in coral as other animals. However, the data do not warrant many of the paper's strong conclusions regarding the overall conservation of the UPRmt in corals and its putative role in protecting against bleaching. This pitfall can be improved through a combination of softening the conclusions along with some additional computational experiments as discussed below.

Author's Response to Decision Letter for (RSPB-2019-0470.R0)

See Appendix B.

RSPB-2019-0470.R1 (Revision)

Review form: Reviewer 1

Recommendation

Accept as is

Scientific importance: Is the manuscript an original and important contribution to its field?

Good

General interest: Is the paper of sufficient general interest?

Acceptable

Quality of the paper: Is the overall quality of the paper suitable?

Good

Is the length of the paper justified?

Yes

Should the paper be seen by a specialist statistical reviewer?

No

Do you have any concerns about statistical analyses in this paper? If so, please specify them explicitly in your report.

No

It is a condition of publication that authors make their supporting data, code and materials available - either as supplementary material or hosted in an external repository. Please rate, if applicable, the supporting data on the following criteria.

Is it accessible?

Yes

Is it clear?

Yes

Is it adequate?

Yes

Do you have any ethical concerns with this paper?

No

Comments to the Author

The authors have addressed my concerns and I support publication of the manuscript.

Review form: Reviewer 2

Recommendation

Accept as is

Scientific importance: Is the manuscript an original and important contribution to its field?

Good

General interest: Is the paper of sufficient general interest?

Acceptable

Quality of the paper: Is the overall quality of the paper suitable?

Good

Is the length of the paper justified?

Yes

Should the paper be seen by a specialist statistical reviewer?

No

Do you have any concerns about statistical analyses in this paper? If so, please specify them explicitly in your report.

No

It is a condition of publication that authors make their supporting data, code and materials available - either as supplementary material or hosted in an external repository. Please rate, if applicable, the supporting data on the following criteria.

Is it accessible?

Yes

Is it clear?

Yes

Is it adequate?

Yes

Do you have any ethical concerns with this paper?

Yes

Comments to the Author

The reviewers have addressed all of my concerns.

Decision letter (RSPB-2019-0470.R1)

03-Jun-2019

Dear Mr Dimos

I am pleased to inform you that your manuscript entitled "Uncovering a Mitochondrial Unfolded Protein Response in Corals and its Role in Adapting to a Changing World" has been accepted for publication in Proceedings B.

Open Access

Paper charges

Sincerely,

Dr Daniel Costa

Appendix A

The manuscript's main ideas and general focus are sound. Broadly, it has been a long-standing goal in molecular evolution to reconstruct the evolutionary progression of major signaling pathways, including the unfolded protein response. Specifically, it has been a major focus in the field of cnidarian-*Symbiodinium* symbiosis to understand the genes and pathways involved in this symbiosis and its ultimate break down upon stress. The researchers take a strong molecular approach in both coral and a model system to understand the evolution of the mitochondria unfolded protein response. In general, coral reef research would be advanced by more studies like this one.

Towards this end, the authors provide some compelling evidence for a conserved response of an ortholog of ATF5 during both thermal and immune stress. In addition, the authors provide evidence that the ATF5-downstream response may be similar to what occurs in bilateral metazoans. It is my general feeling that the data presented suggest that ATF5 might function similarly in coral as other animals. However, the data do not warrant many of the paper's strong conclusions regarding the overall conservation of the UPR^{mt} in corals and its putative role in protecting against bleaching. This pitfall can be improved through a combination of softening the conclusions along with some additional computational experiments as discussed below.

Major Comments:

The authors provide evidence that overexpressing *O. faveolata*'s ATF5 ortholog rescues a mitochondrial-stress-responsive-reporter gene's activity in a ATFS-1 mutant worm. Throughout the text, the authors highlight that this genetic rescue is 'impressive' and that it is the first time this type of experiment has been done with a coral gene (For example: Line 18, 225, 226). However, ortholog replacement studies, like this one, have been conducted successfully for ~30 years between very divergent organisms, such as between yeast and humans. Researchers, familiar with this robust literature, will not be surprised by the finding in this paper. Therefore, I recommend removing these types of statements and focusing on how specifically the ortholog replacement experiments suggests a putative role for ATF5 in coral (See Line 253 comment).

Line 24: "which will likely promote the ability of reef-building corals to survive..." Is there evidence for this? It is not clear to me, based on the data presented, how ATF5 would help corals evolve to increasing environmental stressors. It seems just as likely that any of the other stress-response pathways could be key to coral evolution to a changing climate.

Line 31: To my knowledge, none of these papers (14-16) show that coral produced antioxidants or chaperones are protective from bleaching. Because these papers are based on correlational

data or indirect experiments, please add that these genes are hypothesized to be protective to avoid confusion in the literature. The same point can be made to Line 270-273.

Line 45 and 233: When did ATF5 evolve? Are there non-animal outgroups that have a ATF5 ortholog? Knowing this would help the reader understand the implications of the apparent functional conservation of ATF5 in coral.

Line 56: It is not clear exactly how conserved the UPR^{mt} pathway is in coral. While it is clear that coral ATF5 overexpression can activate the HSP60 promoter in worms, the authors use this to extrapolate that the pathway is largely conserved. However, little effort is made to compare their putative downstream UPR^{mt} responding genes in coral to what is known from model systems. Such analysis would make claims about the existence of a conserved pathway stronger.

Line 98: Elaborating this methods section would be helpful to the reader that is not familiar with worm genetics.

Line 135: Does this criteria for picking target genes actually enrich for genes having ATF5-binding site in their promoters? I am worried that the logic is circular. Looking for known genes in the UTR^{mt} within the module, then finding binding site in their promoters might be not surprising. To help tease apart this, is the module as a whole enriched for genes with ATF5-binding sites compared to the genome as a whole?

Line 181: What are the levels of downregulation with the various RNAi constructs?

Line 190: Are there any other phenotypes associated with the *atfs-1* mutant that the coral gene is able to rescue? This would help support the claim that the coral ATF5 is functional conserved.

Line 190: Does heat activate the UPR^{mt} in *C. elegans* as it is suggested it does in coral (Figure 3)? Specifically, the HSP60 reporter line? Why doesn't the heat used for activating the hsp-16pr::cATF5 transgene launch a UPR^{mt} response (Figure 1D, Control)?

Line 202: Define how you identified target genes in the results to help the reader. Also, can you please give precise numbers here? For example, X of Y of our target genes had a predicted binding motif.

Line 202: What is this significance threshold testing?

Line 213: What test does this P value result from?

Line 213: Are all of these genes used in qPCR known targets of ATF5 or members of the UPR^{mt} in other organisms?

Line 253: It is interesting that the MTS might be functional in *C. elegans* despite the apparent sequence divergence. Can you elaborate on how the *atp-1* and *spg-7* RNAi rescue might suggest a mechanism of ATF5 regulation in the discussion and regarding the model?

Figure 2: Renormalizing the expression of ATF5 by its expression in the control condition like in Figure 3 would help the reader understand the magnitude of upregulation in response to LPS treatment. In addition, it looks like the genes down-regulated in the module (Figure 2B) are mostly driven by replicate LPS-5. Does the variation in ATF5 expression across the replicates correlate with this pattern?

Minor Comments:

Line 52: I am confused by what is meant by “UPR^{mt} governs many of the elements.” Please clarify.

Line 59: “Demonstrate the importance.” Please soften the language because a gene can respond to stress, but it may not be important or required.

Line 133: promoter spelling

Line 139: Genome source and version ref.

Line 168: remove first “ct” from ct delta delta ct.

Figure 1D: The “transgenic” and “non-trangenic” labels are confusing. Aren’t all these lines transgenic?

Line 202: “identify human ATF5 binding motifs” add human

Line 220: “uncover members of a putative UPR^{mt} pathway in ...” add members of a putative UPR^{mt} pathway

Line 221: "Overall animal evolution" There is little comparison of how conserved the pathway is between corals and other animals in this paper. I would suggest removing if pathway-wide analyses are not done.

Line 249: "likewise functions during immune challenge" I would replace the word "functions" with "is upregulated," because it is still unclear if or how that upregulation leads to function.

Appendix B

Reviewer(s)' Comments to Author:

Referee: 1

Comments to the Author(s)

This is a truly trailblazing study, the first one to go all the way to validating the presumed molecular mechanism in corals. It is crystal clear, well written, and combines multiple lines of evidence including rescue of the loss of function mutation in *C.elegans* by a transgenically expressed coral gene.

We thank the reviewer for the enthusiasm relating to our submitted manuscript.

Major concerns:

Search for transcription binding sites for ATF5 was limited to “likely” genes. Would a search among “unlikely” or just randomly picked genes be significantly less successful? Please demonstrate that, otherwise the argument is incomplete.

Thank you for this suggestion. We have since performed a more systematic analysis to address this concern. We scanned the regulatory region of all of our likely genes and found predicted binding sites in 22.4% of the genes. We also scanned the regulatory region of unlikely genes and found predicted binding sites in 3.34%, which is a significant enrichment ($p < 2.2 \times 10^{-16}$) using a fisher's exact test. Our “likely” genes were those identified by the WGCNA analysis, and our “unlikely” or random genes were pulled from a large WGCNA module which had an insignificant relationship with Of-ATF5 ($R = -0.147$, $p = 0.727$) and contained a large number of contigs (1676). We included this new analysis in the updated version of our manuscript with updated methodology in lines 134-147, and results in lines 214-219.

Unsigned WGCNA was performed, meaning that modules would include both positively and negatively correlated genes (as is clear from Fig. 2B). Is this choice justified for ATF5? It would be a reasonable choice if ATF5 could both induce and repress expression of target genes; but since it is mainly inducing (as seems to be the case), signed WGCNA analysis seems more biologically appropriate. Did you try that? (that said, finding of a highly ATF5-correlated WGCNA module with strong enrichment for mitochondrial processes is extremely neat!)

We performed unsigned analysis since ATFS-1 acts as both a transcriptional activator and repressor in *C. elegans* (Nargund, 2012) (Nargund, 2015) and we did want to not bias our results by only searching for positively regulated genes. We did however repeat this analysis using a signed approach and found similar results. In this subsequent analysis we found a module that was highly correlated with expression of Of-ATF5 which contained 881 of the 941 originally identified genes (93.6%) found by our unsigned analysis. GO analysis of the genes recovered from the signed approach likewise revealed enrichment of mitochondrial terms.

Fig 3B contains only 4 points, and 3F – only 3 points of the original 5 (missing one or two less-induced points). Why? Please do show them if you have data.

Unfortunately, we were unable to amplify mtSOD or TIMM-23 by qPCR in all of our samples. We subsequently re-synthesized cDNA and attempted this qPCR again. We were able to get one additional sample to amplify using our TIMM-23 primers but the remaining sample failed to amplify following multiple attempts and was therefore excluded from the analysis. We added this explanation to our results section in lines 230-232.

Does Fig 3 show ALL the genes that were tested for correlation with ATF5?

No, we tested an additional gene: ISCa, a FeS cluster assembly protein which was identified by our WGCNA analysis. We were unsuccessful in our attempts to amplify the ISCa gene with our primers and did not pursue this gene further. This was also re-worded in the text to reflect that five of our investigated target genes had positively correlated expression, instead of all in line 227.

L231-235: Phylogenetic analysis of ATF5 is mentioned here for the first time. Please add this to the Results - and include a figure!

Thank you for this suggestion. We moved the results of our phylogenetic analysis from the discussion into our results lines 187-192 and moved our gene tree which was in our supplementals into the main figure 1 to address this concern.

L255: "UPR^{mt} of *O. faveolata* might function similarly to the UPR^{mt} of *C. elegans* by promoting immune competence (figure 4). Overall, this indicates that the role mitochondria as purveyors of innate immunity (57, 58) and mediators of cell death has a deep evolutionary origin and functions in the immunology of corals" – there is no notion of immune competence on Fig.4, and, as far as I can see, there is no data here to suggest that function Of-ATF5 affects coral immunity (although it does respond to the immune challenge).

(on a similar note) L237-240: "We conclude that Of-ATF5 likely mediates a UPR^{mt} in *O. faveolata* which is similar to the response mediated by ATFS-1 in *C. elegans*, by regulating the expression of genes that are involved in protein homeostasis (54), detoxification of damaging free radicals (20), and innate immunity (23)." – technically, what is demonstrated here is that it Of-ATF5 can regulate a heat shock protein in *C.elegans*, is correlated with mitochondrial genes and genes involved in cell death (by GO analysis in a coral), and with an antioxidant gene mtSOD; but there seems to be no direct evidence for its involvement in immunity.

Thank you for these comments. Given that the response mediated of Of-ATF5 and ATFS-1 are so similar we extrapolated this to suggest that the immune promoting abilities of the UPR^{mt} are likewise conserved. We agree that this an overinterpretation of our data and

amended our original statement to suggest that “future investigations should explore if the immune promoting abilities of the UPR^{mt} are conserved in corals” in lines 269-271.

To be honest I find the whole discussion section about “The role of the UPR^{mt} in coral disease” premature. The paper nicely establishes that corals possess UPR^{mt} response, but it remains to be seen what role does it actually play in disease and immunity. It responds to immune challenge (which is nicely shown), but this could be a response to general cellular stress rather than a specific immunity mechanism (for example, under any stress ribosomal genes tend to get down-regulated, reflecting decrease in growth).

We agree with the reviewer and have reduced the tone in the discussion. We believe that we have likewise addressed this comment in response to the previous comment by amending our original statement to “the UPR^{mt} of *O. faveolata* might function similarly to the UPR^{mt} of *C. elegans* by promoting mitochondrial recovery during immune challenge.” In lines 267-269.

Minor things:

Figure 3 legend – please spell out the names of proteins correlated with ATF5 here.

We have made this amendment in lines 327-332.

Figure 4 is a general summary of ATF5 function rather than a conclusion of this study (if we just remove “Of-” from the label of the protein), so maybe make it Fig 1 and refer to it in intro?

We believe that this figure is a summary of our findings and not a review of human ATF5. To our knowledge, the role of human ATF5 during heat stress has not been investigated and our thus our findings present a conclusion that is unique to Of-ATF5. Also, while the mechanism of Hs-ATF5 regulation involves organelle partitioning, it is still unknown for Of-ATF5 prompting us to include the statement of “unknown sensing mechanism” in our model. We believe that these differences justify inclusion of the model in order to promote other investigations and studies into this system.

Fig. 1A: I am not clear why Of-ATF5 has a long gap near the N-terminus after position 9. The alignment would be perfectly fine without this gap.

We included this extra gap to include the entire bZIP domain but will remove it in a revised figure 1 for clarity.

L304: Fig 2 legend: “Fraction indicates number of genes with values exceeding significance threshold in each term” – if the GO_MWU was run correctly for WGCNA modules, this fraction

should be (number of genes from this GO category assigned to the module) / (total number of genes in this GO category). Please check!

Thank you for noticing this error. We ran our GO_MWU analysis with the option for unsigned WGCNA analysis and failed to update the methodology from the standard workflow. The numbers that we show do indeed represent the number of genes containing that term in our module, and this change can be found in lines 323-324.

Fig. 2B: is in this heatmap, please indicate with an arrow the row corresponding to Of-ATF5.

We have updated the heatmap with this addition and the figure legend in line 320.

(several places in text) "obscured" obscure

We have updated the text with this amendment.

L122: "Rlog transformed contigs" Rlog transformed counts

This was adjusted in line 127.

L125-127: The last sentence in the WGCNA description creates an impression that the three named traits were used to guide the coexpression network construction, which is not how it works. I suggest rephrasing: "The behavior of the identified coexpression network modules was investigated with respect to three traits: ...".

We apologize if our wording may have been misleading, this was corrected in lines 131-132. WGCNA uses an unguided approach (with respect to external traits to create modules).

L133: "prompter" promoter

This typo was corrected.

L155: "either held at either" either held at

This error was corrected.

L198: "figure 3B" - I am guessing it should be Fig. 2B.

This is correct and was changed

L268: "We therefore developed model of a mitochondrial stress-based mechanism of dysfunction (figure 4)" – as I mentioned previously, this figure is in fact the summary of prior knowledge about Hs-ATF5 function in which "Hs" is substituted for "Of", not the "model"

substantiated by the results of this study. For example, this work presents no direct evidence of nuclear translocation, of the involvement of ROS, loss of proteostasis, or loss of membrane potential in affecting Of-ATF5 function. There is also no evidence that ATF5 function actually leads to mitochondrial recovery.

We believe that our previous comment addresses this concern. While we do not provide direct evidence of mitochondrial recovery in a coral we rely upon our transcriptional data and knowledge about the UPR^{mt} from model systems to make this statement. Our transgenic *C. elegans* experiments suggest that Of-ATF5 can be imported into the nucleus since it can activate transcription of the *hsp-60::GFP* reporter in the absence of ATFS-1. Nonetheless, we agree with the reviewer and will change the wording in this figure to reflect the hypothetical nature of this model in lines 267 and 283. However, we assert that placing our findings in a graphical context of a working model is valuable to assist readers in understanding this cellular pathway and thus should be included.

Referee: 2

Comments to the Author(s)

This manuscript describes a mitochondrial unfolded protein response in corals by describing a transcription factor ATF5-like homolog in corals, examining its function in a transgenic system in *c. elegans*, and performing a bioinformatics analysis linking this gene with expression of other genes during an immune challenge and temperature stress response.

Overall I feel that the work was capably carried out. I appreciate the collaboration between coral biologists and geneticists from other fields who are bringing new perspectives and techniques to the field of coral biology. I do have some concerns that I feel can be easily addressed in a revision.

First and most importantly, I feel that the authors overplay their hand, especially in the rationale for carrying out the work. At this point, there are several dozen reviews on coral bleaching, indeed 3 within the past year alone. Although we do not have every detail on the mechanisms of bleaching, by any stretch, we do know quite a bit at this point. So wording in lines 32-35 and lines 259-283 should be modified – sure we don't know the level of detail on par with well-developed model systems, we nonetheless know a lot. The UPR is an interesting addition to the repertoire of bleaching mechanisms but it is not particularly surprising or unexpected. It slots right in with the variety of other stress/immune/ROS responses that have been studied and described for years. I feel that the authors need to tone down both the rationale and the conclusions section of the paper and provide the proper context for where UPR fits in the story.

We thank the reviewer for their insightful comments.

We have made alterations to address these concerns by amending our wording in lines 32-35 to reflect the extensive field of coral bleaching, while proposing that additional aspects to coral bleaching may be resolved using cellular biology.

In addition, we made changes to our discussion to imply that the UPR^{mt} is not the only relevant cell mechanism in coral's response to environmental stress by amending our original statement to "may be a contributing pathway" in line 289.

There are areas in the manuscript where I do not feel that the coral biologists have added enough of their content knowledge. This is especially notable regarding the discussion of ATF5 homologies and trees (Fig S1 and lines 229-231). It is now well recognized that the Ecdysozoa are outgroups in the phylogenomics of the Metazoa, see for example Miller et al Genome Biology 2007. Genes and genomes of early diverging metazoans more closely resemble those Deuterostomes including chordates than the Ecdysozoans. So it is not at all surprising that coral ATF5 is closer to vertebrates than *C. elegans*. This should be modified in the discussion.

We appreciate the insight provided by the reviewer. While the genome of early diverging metazoans more closely resembles dueterostomes, *C. elegans* pathways like the UPR^{mt} are commonly studied to gain insights in order to extrapolate to man. This is not commonly applied to corals which is why we included language like "surprising". We agree with the review and have amended statements to remove words like "surprising".

While I understand that length limits sometimes constrain explanations of methods and results, I feel in places that there is an absence of information. Figure 1D is a nice qualitative measure of knockdown and rescue, but where are quantitative data backing these findings up? The methods for Figure 2A are missing. What is rlog? QPCR? Other method? What statistics were performed? Explain. I have similar comment for the LPS experiment, lines 105-108. Could someone repeat this experiment using this description? How did you arrive at this concentration of LPS and the 4 hour incubation time. A supplementary figure and methods could explain pilot experiments that helped you arrive at this end point. What is the vehicle and at what concentration?

Thank you for this comment. We have included a quantification of the fluorescence from our genetic rescue experiment. The experimental methods for figure 2A have been previously published in this journal and we thought that the article by Fuess et al. (2017) describes the experiment sufficiently well to reproduce it. Rlog is a common way to normalize transcriptome data when performing individual gene comparisons, and is common in transcriptomics use. We do explain how this normalization was created as well as the statistics used in our comparison. "Of-ATF5 expression level from the rlog normalization was used in an unpaired t-test (n=4 per group)." in lines 121-122.

Referee: 3

Comments to the Author(s)

The manuscript's main ideas and general focus are sound. Broadly, it has been a long-standing goal in molecular evolution to reconstruct the evolutionary progression of major signaling pathways, including the unfolded protein response. Specifically, it has been a major focus in the field of cnidarian-*Symbiodinium* symbiosis to understand the genes and pathways involved in this symbiosis and its ultimate break down upon stress. The researchers take a strong molecular approach in both coral and a model system to understand the evolution of the mitochondria unfolded protein response. In general, coral reef research would be advanced by more studies like this one.

Towards this end, the authors provide some compelling evidence for a conserved response of an ortholog of ATF5 during both thermal and immune stress. In addition, the authors provide evidence that the ATF5-downstream response may be similar to what occurs in bilateral metazoans. It is my general feeling that the data presented suggest that ATF5 might function similarly in coral as other animals. However, the data do not warrant many of the paper's strong conclusions regarding the overall conservation of the UPR^{mt} in corals and its putative role in protecting against bleaching. This pitfall can be improved through a combination of softening the conclusions along with some additional computational experiments as discussed below.

We thank the reviewer for the enthusiasm expressed towards our manuscript as well as their helpful critiques.

Major Comments:

The authors provide evidence that overexpressing *O. faveolata*'s ATF5 ortholog rescues a mitochondrial-stress-responsive-reporter gene's activity in a ATFS-1 mutant worm. Throughout the text, the authors highlight that this genetic rescue is 'impressive' and that it is the first time this type of experiment has been done with a coral gene (For example: Line 18, 225, 226). However, ortholog replacement studies, like this one, have been conducted successfully for ~30 years between very divergent organisms, such as between yeast and humans. Researchers, familiar with this robust literature, will not be surprised by the finding in this paper. Therefore, I recommend removing these types of statements and focusing on how specifically the ortholog replacement experiments suggests a putative role for ATF5 in coral (See Line 253 comment).

While ortholog replacement studies are common place in a variety of model organisms, we feel our claims about novelty were not unjustified as we are unaware of previous attempts to perform ortholog replacement studies using coral genes. However, we have removed these statements from the text in order to reduce the tone of the finding's novelty.

Line 24: "which will likely promote the ability of reef-building corals to survive..." Is there evidence for this? It is not clear to me, based on the data presented, how ATF5 would help corals evolve to increasing environmental stressors. It seems just as likely that any of the other stress-response pathways could be key to coral evolution to a changing climate.

The reviewer is correct that we have present no direct evidence that this pathway will allow corals to adjust to increasingly stressful environments. However, we make this claim in order to provide context as our results builds upon substantial knowledge into the mechanisms of stress tolerance in corals. Doing so allows us to speculate that the previously suggested mechanisms of acclimation in corals may be mediated by the pathway we present. We did amend our original statement to “may” line 23 in order to place our findings in the proper context.

Line 31: To my knowledge, none of these papers (14-16) show that coral produced antioxidants or chaperones are protective from bleaching. Because these papers are based on correlational data or indirect experiments, please add that these genes are hypothesized to be protective to avoid confusion in the literature. The same point can be made to Line 270-273.

This comment is well founded and the suggested edits were made. Similar issues were raised by reviewer 2 and we believe our amendment to the text adequately addresses this concern by saying “have been suggested to mediate a protective response” in line 31 and line 287&288.

Line 45 and 233: When did ATF5 evolve? Are there non-animal outgroups that have a ATF5 ortholog? Knowing this would help the reader understand the implications of the apparent functional conservation of ATF5 in coral.

We thank the reviewer for this comment. We probed for ATF5 homologs in the two most basal animal lineages, sponges and Ctenophores. The sponge *Amphimedon queenslandica*, has a weakly predicted homolog to ATF5 and no clear homolog was identified in the Ctenophore *Mnemiopsis leidyi*. As an animal outgroup we investigated the protist choanoflagellates *Salpingoeca rosetta* and *Monosiga brevicollis* for ATF5 homologs. We found several bZip-containing proteins in these species, but no clear homologs of ATF5. Despite this several proteins possessed ATF4/5-like bZip domains. We also looked into the the model plant species *Arabidopsis thaliana* where we were unable to detect any homologous sequences to ATF5. Due to the low level of similarity, like what we observed in Of-ATF5 (Which necessitated ortholog replacement studies to validate the conservation) it is too premature to make comments in the main text as to role of any other putative ATF5 homologs regarding involvement in a UPR^{mt}.

Line 56: It is not clear exactly how conserved the UPR^{mt} pathway is in coral. While it is clear that coral ATF5 overexpression can activate the HSP60 promoter in worms, the authors use this to extrapolate that the pathway is largely conserved. However, little effort is made to compare their putative downstream UPR^{mt} responding genes in coral to what is known from model systems. Such analysis would make claims about the existence of a conserved pathway stronger.

We feel that our computational analysis provided in figure 2 supports that the overall functions of this pathway are conserved from *C. elegans* to *O. faveolata*, as the gene

ontology terms associated with our analysis are consistent with the known role of the UPR^{mt} in the worm (Nargund, 2012) (Nargund, 2015).

Line 98: Elaborating this methods section would be helpful to the reader that is not familiar with worm genetics.

We agree and have updated the methods section appropriately which can be found in lines 97-101.

Line 135: Does this criteria for picking target genes actually enrich for genes having ATF5-binding site in their promoters? I am worried that the logic is circular. Looking for known genes in the UTR^{mt} within the module, then finding binding site in their promoters might be not surprising. To help tease apart this, is the module as a whole enriched for genes with ATF5-binding sites compared to the genome as a whole?

Reviewer 1 raised similar concerns and we preformed subsequent more systematic analysis by comparing the presence of Of-ATF5 binding motifs within the regulatory region of our putative UPR^{mt} genes compared to a set of “random” genes. We performed this analysis with “random genes” as the program used to identify binding motifs (fimo) specifically mentions in the manual that it is not well-suited for whole-genome searches. In this subsequent analysis we identified significant enrichment of ATF5 binding motifs within our identified putative UPR^{mt} genes, compared to a set of random genes and found significant enrichment based upon a fisher’s exact test ($p < 2.2 \times 10^{-16}$). These amendments were made in lines 134-147 and lines 214-219.

Line 181: What are the levels of downregulation with the various RNAi constructs?

We have since performed a qPCR for both *atp-2* and *spg-7* in worms raised on the respective RNAi clones and this has been included in supplementary figure 2 and in our methods line 101. In summary our *atp2* and *spg-7* RNAi clones depleted expression of the respective gene by 87% and 77%, respectively.

Line 190: Are there any other phenotypes associated with the *atfs-1* mutant that the coral gene is able to rescue? This would help support the claim that the coral ATF5 is functional conserved.

This is an excellent suggestion. However, while the expression of Of-ATF5 rescues the activation of the UPR^{mt} reporter, expression of high-copy arrays for proteins from both *C. elegans* or its orthologs often leads to developmental issues, due to the overexpressed nature of the protein. This is true for proteins belonging to broad list of function categories in *C. elegans*. This precludes any sort of examination of physiological or developmental attributes.

Line 190: Does heat activate the UPR^{mt} in *C. elegans* as it is suggested it does in coral (Figure 3)? Specifically, the HSP60 reporter line? Why doesn’t the heat used for activating the hsp-16pr::cATF5 transgene launch a UPR^{mt} response (Figure 1D, Control)?

This is an interesting question. While the effect of temperature stress on the UPR^{mt} has not been extensively studied, we have observed activation of *hsp-60::GFP* as well as another UPR^{mt} reporter (the mitochondrial chaperone mtHSP70 homolog reporter *hsp-6::GFP*) at elevated temperatures. We have included this as supplementary figure panel (Suppl. Fig. 1) and in our results lines 224-225. We were not able to investigate this property with our Of-ATF5 worm due to concerns about the nature of multi-copy transgenes addressed in the previous comment.

Line 202: Define how you identified target genes in the results to help the reader. Also, can you please give precise numbers here? For example, X of Y of our target genes had a predicted binding motif.

This is a valuable suggestion and we have amended our original analysis to a more systematic one. Of our 941 genes identified through our network analysis, we were able to confidently assign 807 of these to annotated regions of the *O. faveolata* genome and identified putative binding sites in 181 of the genes and show these results in an updated supplemental table 2 in the text in the lines mentioned in our previous comment.

Line 202: What is this significance threshold testing?

The program that we used to perform this analysis was Find Individual Motif Occurrence (FIMO). This program computes a log-likelihood ratio score for a given nucleotide at each position of a sequence, and then converts this score to a P-value, using a null model generated using the genomic nucleotide background, and under standard parameters uses a significance threshold of 1e-4.

Line 213: What test does this P value result from?

Log-likelihood ratio score tested against a null-model. See previous comment.

Line 213: Are all of these genes used in qPCR known targets of ATF5 or members of the UPR^{mt} in other organisms?

Of our 5 tested genes 4 are known components of the *C. elegans* UPR^{mt}. HSP-60 and mtHSP70 are two mitochondrial chaperone genes that are regulated by ATF5 and ATFS-1 in mammals and *C. elegans*, respectively. The *C. elegans* homologs of TIMM-23 and mtSOD were identified as part of the *C. elegans* UPR^{mt} in Nargund et al. 2012 and this information was added to our methods section lines 170-172. The homolog of IMP-2 is not currently known to function in the UPR^{mt} of *C. elegans* but its function as a mitochondrial protease and well supported correlation with Of-ATF5 identified by our network analysis lead us to investigate if this gene similarly has a correlated expression pattern with Of-ATF5 during temperature stress which is now included in line 173.

Line 253: It is interesting that the MTS might be functional in *C. elegans* despite the apparent sequence divergence. Can you elaborate on how the *atp-1* and *spg-7* RNAi rescue might suggest a mechanism of ATF5 regulation in the discussion and regarding the model?

We believe that our results using RNAi against *atp-2* or *spg-7* indicate the Of-ATF5 is likely regulated by organelle partitioning like ATFS-1/Hs-ATF5 which we indicate in the text. In this context, mitochondrial dysfunction reduces mitochondrial import efficiency allowing nuclear import of the transcription factor. However, we decided not to extend this to the model as we have no direct evidence of Of-ATF5 trafficking to the mitochondria during homeostasis.

Figure 2: Renormalizing the expression of ATF5 by its expression in the control condition like in Figure 3 would help the reader understand the magnitude of upregulation in response to LPS treatment. In addition, it looks like the genes down-regulated in the module (Figure 2B) are mostly driven by replicate LPS-5. Does the variation in ATF5 expression across the replicates correlate with this pattern?

We had chosen to use Rlog for our analysis since it is a widely used method to normalize sequencing data for gene by gene comparison. The variation in Of-ATF5 expression is very low across samples, and the expression in replicate LPS-5 is not an outlier.

Minor Comments:

Line 52: I am confused by what is meant by “UPR^{mt} governs many of the elements.” Please clarify.

We used the term “governs” in proxy of regulates and this was changed to “regulates” to help add clarity in the revised version of our manuscript line 51.

Line 59: “Demonstrate the importance.” Please soften the language because a gene can respond to stress, but it may not be important or required.

We agree and reduced the tone of the sentence to state: “We also demonstrate that due to its increased expression, Of-ATF5 may function in coral during both immune challenge and heat stress” in lines 58-59.

Line 133: promoter spelling

This was corrected

Line 139: Genome source and version ref.

We have since added the Genome and source and version ref in the revised version of our manuscript in the methods line 139.

Line 168: remove first “ct” from ct delta delta ct.

This was amended in line 175.

Figure 1D: The “transgenic” and “non-trangenic” labels are confusing. Aren’t all these lines transgenic?

This is correct, all of the lines are transgenic. We have updated the figure by labelling the strains as “wild-type”, “*atfs-1(tm4525)*”, and “*atfs-1(tm4525)* + OfATF5”, and moved the label “*hsp-60pr::GFP*” to the panel of the image in fluorescent green for clarity.

Line 202: “identify human ATF5 binding motifs” add human

This was corrected and is found in line 216.

Line 220: “uncover members of a putative UPR^{mt} pathway in ...” add members of a putative UPR^{mt} pathway

This was corrected and is found in line 239 in our revised manuscript.

Line 221: “Overall animal evolution” There is little comparison of how conserved the pathway is between corals and other animals in this paper. I would suggest removing if pathway-wide analyses are not done.

This over-arching statement was removed to more accurately reflect our findings.

Line 249: “likewise functions during immune challenge” I would replace the word “functions” with “is upregulated,” because it is still unclear if or how that upregulation leads to function.

This change was made to reflect that Of-ATF5 is upregulated during immune challenge and is found in line 262 in our revised manuscript.